# Comparative Morphological Evaluation of Young Women’s Breast-Bra Reshaping by Different Bra Cups

**DOI:** 10.3390/ijerph20053856

**Published:** 2023-02-21

**Authors:** Zejun Zhong, Beibei Zhang, Yupeng Hu, Lingling Zhang, Bingfei Gu, Yue Sun

**Affiliations:** 1Fashion College, Zhejiang Sci-Tech University, Hangzhou 310018, China; 2Clothing Engineering Research Center of Zhejiang Province, Hangzhou 310018, China; 3Key Laboratory of Silk Culture Heritage and Products Design Digital Technology, Ministry of Culture and Tourism, Hangzhou 310018, China

**Keywords:** breast-bra shaping, morphological parameters, shape variations, predicted models

## Abstract

Female breasts are regarded as a factor reflecting women’s morphological beauty. An appropriate bra can fulfill aesthetic needs, thus boosting self-esteem. This study proposed a method to analyze young women’s breast-bra morphological variations between two identical bras with different bra cup thicknesses. The 3D surface scan data of 129 female students who were braless and wore a thin bra (13 mm) and a thick bra (23 mm) were analyzed. Integral sections of the breasts and bra were cut at a fixed thickness of 10 mm, and slice maps were derived. Morphological parameters were extracted in braless and the two bra conditions. The variations in breast-bra shape caused by different thicknesses of bra cups were evaluated by quantifying breast ptosis, gathering, and breast slice area. The results showed that the thin bra lifted the breasts by 2.16 cm, whereas the thick bra decreased breast separation, gathering the breasts and moving them 2.15 cm laterally towards the center of the chest wall. Moreover, prediction models constructed using the critical morphological parameters were used to characterize breast-bra shape after wearing the provided bras. The findings lay the groundwork for quantifying the breast-bra shape variation caused by different bra cup thicknesses, allowing young females to choose optimally fitting bras to achieve their desired breast aesthetics.

## 1. Introduction

Breasts are regarded as a factor reflecting women’s morphological beauty and are also a focus of women’s health attention [1,2,3,4]. Round, full, properly proportioned breasts are widely regarded as a manifestation of young feminine charm in different cultural backgrounds [5]. However, the breast shapes of female bodies from different regions, at different ages, or with different living habits are quite different [6,7,8]. Breasts, especially larger breasts, tend to sag when a bra is not worn for a long time, due to the extensive stretch of soft tissues and ligaments [9]. Breast separation and ptosis may lead to health problems such as mastalgia [10], shoulder and back pain [11], and posture alteration due to the downward pull of breasts [12]. Dissatisfaction with one’s appearance can lead to mental issues such as body-related anxiety and a sense of inferiority [13]. Therefore, in daily life, a bra is an essential piece of clothing for women to maintain an ideal shape or support the breasts. A well-fitted bra can play an important role in reshaping the body into a desirable silhouette. In contrast, a poorly fitted bra can lack this reshaping effect due to empty cups or excessive pressure [14,15,16]. With increasing emphasis on aesthetic fashion, women are willing to pay more attention to bras’ function and morphological change in the breasts when wearing bras, which makes the correct bra sizing and the design of desired functions critical design issues in the product development stage of bras [17].

Breast morphology is the basis for bra design and wearing appearance, especially for a well-fitting bra [1,18]. An appropriate bra is expected to modify the breast shape aesthetically in addition to providing a supportive function. Due to the differences in breast morphology, classification is necessary to provide references for bra design. Liu et al. [19] analyzed 108 measurement items in 267 subjects, obtaining eight categories of breast shape to provide a theoretical foundation for choosing the most desirable bra size and shape. In previous studies [20,21,22] on breast morphology, although many scholars have introduced angles as measured parameters, some basic body parameters are still lacking, including the shoulder oblique angle and the angles formed by the shape contour of the breast. Zhang et al. [23] measured six parameters, including breast volume and angles extracted from breast contours, in 158 young female college students, classifying nine types based on local breast features and overall shapes. These studies on breast morphology can provide theoretical support and data references for bra design and pattern production. Based on these refined classifications of breast shape, the design and production of bras will be more in line with the female breast shape, thus improving fit and wearing comfort.

The pattern design of bras is based on breast shape, and women with different breast shapes need to wear bras with different cup shapes to obtain reasonable protection and maintain an aesthetic shape [14,15,16]. International Standard bra sizes use the under-bust girth to represent band size with 5 cm intervals, and use the difference between the bust and under-bust girths to represent breast shape with 2.5 cm intervals. For Chinese Standard bra sizes, such as FZ/T 73012–2017, the detailed principles are basically similar to those of Europe, America, and other regions [24,25]. However, existing breast standards cannot reflect the 3D morphological differences in female breasts [23]. Bra cups, which provide overall protection as well as reshaping the breasts, are usually designed according to the morphology of the breasts [26]. However, due to variation in bra cup shape and variation in adipose and glandular tissue, the appearance of the breast-bra shape may vary greatly when wearing different kinds of bras [27]. Bra cups with various thicknesses can redistribute the soft tissue of the breasts to different degrees [28]. Even for the same person, identical bras with different cup thicknesses may result in different breast-bra shapes. Therefore, in order to explore bras that produce the ideal shape in women with various types of breasts, the relationship between cup thickness and ultimate breast-bra shape urgently needs to be explored.

To analyze the changes in breast-bra shape associated with different bra cup thicknesses, this study proposes a method for comparing 3D differences in breast-bra morphology among three states, i.e., braless, when wearing bras with thin cups, and when wearing bras with thick cups. The measurements were conducted based on the classifications of breast shape obtained from our previous research [23]. This study will provide a foundation for the quantitative expression of shape variations caused by bras with different thicknesses, thus helping young women select bras that fulfill their aesthetic needs.

## 2. Methodology

### 2.1. Participants

This experiment was approved by the University Ethics Committee and all participants were provided with a written informed consent sheet. According to ISO 15535:2012, the minimum sample size required for each measurement item is at least 97. Therefore, 129 young females (BMI: 20.34 ± 2.51 kg/m^2^, age: 20.5 ± 1.90) were recruited from Zhejiang Sci-tech University. According to FZ/T 73012–2017, their cup sizes were in the range of B and C. Pregnant women, breast-feeding women, and patients with a history of breast surgery were excluded from the study to eliminate the effect of hormone levels on the breast.

### 2.2. 3D Body Scanning

The 3D point cloud body data were obtained using a [TC]^2^ non-contact body scanner (NX16, America; accuracy: 0.6 mm). According to ISO 20685-1:2018(E), in case the hair affected the accuracy of the collected data around the neck and shoulders, an elastic cap was used to cover the hair. Participants were asked to stand on the fixed footprints and be parallel to each other. Their heads were in the Frankfurt plane, and their eyes were looking straight forward. The upper arms were abducted to hold the auxiliary rods with the elbows straight and the palms facing down. During the scan, participants should breathe quietly and maintain a relaxed shoulder position.

According to online shopping data, 3/4-cup bras with uniform sizes were the most popular in China [29,30]. Therefore, two 3/4-cup bras of uniform size were selected for participants. Information about the provided bras is shown in Table 1. Each participant was scanned under three conditions, i.e., wearing a thin bra (Bra A), wearing a thick bra (Bra B), and without wearing any bra (braless).

The bra-fitting issue of all participants was assessed using professional criteria [31]. During the bra-fitting process, subjects were asked to try on the bra samples under the guidance of a bra-fitting specialist. Firstly, the correct tensions of the under-band and the shoulder straps were adjusted. Secondly, the subject leaned forward so that her breasts could fully fill in the cup to ensure no gap, and other ill-fitting problems such as bulging, wrinkles, digging, or sliding were also checked and avoided. Lastly, the subject raised her hands to check that the bra remained firmly in place without riding up. After finishing all the assessments, further 3D scanning could be continued.

### 2.3. Parameter Determination Based on Breast Slicing

Inspired by topographic and related literature [27,32], breasts can be regarded as mountains to evaluate the elevation changes. A section between the breast base and bust point can be sliced to form a breast slice map, which provides precise measurements for breast surface morphological analysis, especially the 3D changes in breast shape after wearing different kinds of bras. In order to obtain the breast slice map, reference points and lines should be extracted. The definitions of all reference points, lines, and parameters are shown in Table 2.

Step 1: Definition of feature points;

Feature points are the basis and key for determining breast slice location and performing breast morphological analysis. Referring to ISO 7250-1:2017 and related studies on breast morphology [23,33,34], four feature points (i.e., BP, MBP, LBP, and UBP) and four feature lines (i.e., BL, UBL, LML, and L_1_) were determined according to the definitions shown in Table 2. The legend of feature points and lines is shown in Figure 1. Among the four feature points, BP is the most important reference point for breast aesthetic standards and garment pattern making. The other points (MBP, LBP, and UBP) form the bottom breast boundary, which is related to the shape of the bra cup.

Step 2: Automatic positioning of feature points;

Automatic positioning of feature points based on 3D point cloud data is required to automatically slice breast and extract feature parameters. Head, neck, arms, and the part below under-bust of each subject’s 3D point cloud data were all removed by automatically identifying the landmarks, such as the front neck point, axillary point, and UBP. The upper body part was thus obtained for the next round of automatic positioning of four feature points.

Four feature points were first determined according to the global coordinate system fixed to the ground’s surface. The origin of the global coordinate system was the intersectional point of the sagittal plane, coronal plane, and ground surface plane. The x-axis (the sagittal axis) referred to the fore-and-aft direction (pointing forward), the y-axis (the transverse axis) referred to the side-to-side direction (pointing to the right), and the z-axis (the longitudinal axis) referred to the top-to-bottom direction (pointing upwards).

The process of identifying four feature points is as follows:UBP: Firstly, point AP1, the convex breast surface with the maximum x-value, was automatically identified. An AP1-based searching area (SA_1_) was determined by the part of the body surface extracted from four boundary planes (Figure 2a). In order to automatically characterize the UBP for all subjects, the positions of four boundaries were determined after trials to ensure the SA_1_ could cover the under-bust line. The upper boundary was finally set as the horizontal plane through AP1, and the bottom was 10 cm below. The left and right boundaries were set ±5 cm away from the vertical plane through AP1, respectively. Multiple sagittal curves with an interval of 0.5 cm were extracted from the body surface in the searching area. The maximum curvature points in each sagittal curve were extracted, among which UBP was identified as the minimum z-value (Figure 2b);MBP: A-plane and B-plane were defined as horizontal planes passing through the axillary point and UBP, respectively. C-plane was the half-plane between them (Figure 2c). MBP was denoted as the maximum curvature point at the intersection curve (L_1_) between the C-plane and body surface (Figure 2d).LBP: The auxiliary point (AP2) was the maximum curvature point at the half-breast contour section of L_1_, and LBP was the point on L_1_ with a distance from AP2 equal to that between MBP and AP2 (Figure 2d);BP: Another searching area (SA_2_) was determined by the part of the body surface extracted from four boundary planes (Figure 2e). The upper and bottom boundaries were 2 cm away from the horizontal plane through AP1, and the left and right boundaries were the vertical planes through LBP and MBP. BP was the outermost point on the line (LML) passing through points LBP and MBP in SA_2_ (Figure 2f).

Step 3: Coordinate transformation;

The bust cross-sectional curve was extracted after the determination of feature points. The local coordinate system of the bust cross-section curve was set using the minimum bounding rectangle method, as shown in Figure 3. The center of the rectangle was set as the coordinate origin point O. The *x*-axis and *y*-axis were constructed from horizontal and vertical lines through point O, respectively. The right bust was selected to rebuild the new coordinate system. By projecting the line LML into the original *x*-*y* coordinate plane, a line perpendicular to the projection of LML was drawn by passing through point BP, and the intersection point of the line and the original *y*-axis was defined as point O’ which was considered the new origin. Therefore, the line passing through O’ and BP was considered the new *x*-axis, and the new local coordinate system is shown in Figure 3.

Step 4: Breast slicing;

After measuring the point cloud data obtained from the 3D scanner, the minimum spatial distance between any two points was 5 mm. Therefore, the slice thickness was determined as 10 mm to cut the breast-bra part into different slices, as shown in Figure 4a. According to the maximum slice number and the distance between the slice and point O’, the slices were described in different colors, as shown in Figure 4b,c. The color of the slice changed from purple to red as the distance from point O’ increased. Each slice was divided into four quadrants according to the coordinate axis, and then were named S_ij_ (i = 1, 2, …s, stands for slice number; j = 1–4, represents the quadrant). Breast-bra slice maps can be used to conduct quantitative and qualitative analyses of breast morphological changes.

Step 5: Parameter extraction.

Appropriate parameters are fundamental and necessary for morphological analysis. Breast morphological parameters, including angles, area, girths, heights, widths, depths, and ratios, were automatically measured after positioning the feature points and slicing the breast. As shown in Table 2, A_LBM_ are the angle parameters; S_ij_ are area parameters; G_BL_ and G_UBL_ are girth parameters; H, H_BP_, H_UBP_, H_T_, and H_U_ are height parameters; W, W_BB_, W_MBP_, W_LM_, and W_LBP_ are width parameters; D and D_S_ are depth parameters; and R_BH_ and R_BW_ are ratio parameters.

## 3. Results

### 3.1. Test of Outliers

Since there are occasional and systematic errors during the body measuring process, the parameters obtained from the 129 samples need to be preprocessed to eliminate the missing values and outliers, thus strengthening the statistical outcomes [35,36]. After examination and correction, 124 valid samples were finally obtained for further analysis.

The pairwise Wilcoxon rank-sum test was performed to show whether there were significant differences among corresponding parameters under different wearing conditions. The results showed that *p*-values were less than 0.05 for all parameters, indicating a statistically significant difference. All parameters of Bra A, Bra B, and braless would be taken for further analysis to detect the breast-bra morphological variations under the three different conditions.

### 3.2. Statistical Analysis of the Morphological Parameters

To better understand the variations in breast-bra shape caused by different bra thicknesses, three conditions were compared: the improvement degree of breast ptosis and gathering, and the breast slice area. Moreover, the variations with different breast shapes were also analyzed according to our previous research on the classification of breast morphology [23]. The breast shape was classified into three types: 46 samples belonged to the plump oval type, 50 samples belonged to the uniform oval type, and the other 28 samples belonged to the flat circle type.

#### 3.2.1. The Degree of Breast Ptosis

Due to the tendency of female breasts to ptosis as they age, women prefer to wear bras to support breasts by lifting the degree of BP height [37]. The parameters, e.g., H_T_, H_U_, and H_BP_, which could quantify the amount of breast ptosis, were compared among the three conditions. The variations of these three parameters are expressed in Table 3. For H_T_, a higher value was observed after wearing Bra B than after wearing Bra A, whereas H_U_ values showed an opposite trend. In comparison with the braless condition, the H_T_ values in Bra A and Bra B both decreased, but Bra A had a more obvious reduction of 1.92 cm compared with 1 cm for Bra B. The H_U_ value increased, indicating that the distance between the BP point and the UBP increased after wearing a bra, especially for Bra A (1.95 cm for Bra A vs. 1.12 cm for Bra B). Moreover, the average value of H_BP_ observed in Bra A was greater than that in Bra B, indicating that Bra A had a better effect on an upward trend of BP position by lifting 2.16 cm (1.18 cm for Bra B).

In addition, the improvement degree of breast ptosis can also be evaluated by the R_BH_ value. Liang [38] concluded that breasts could be considered as a standard shape if the R_BH_ value was 0.71 ± 0.005. When the R_BH_ value is less than 0.705, it is considered low-breast, and when the R_BH_ value is greater than or equal to 0.715, it is regarded as high-breast. Table 4 shows the distribution of the R_BH_ value for three types among three wearing conditions. As for plump oval type and uniform oval type, most subjects belonged to the low-breast category in braless, the standard-breast category in Bra B, and the high-breast category in Bra A. As for flat the circle type, most subjects belonged to the high-breast category for all three conditions, especially for Bra A.

#### 3.2.2. The Degree of Breast Gathering

Since the breast shape of most women shows a separating appearance under nude state, they prefer to wear bras with gathering functions. The parameters, e.g., W_LBP_, W_MBP_, and W_BB_, which could represent the degree of breast gathering, were compared in this study. The variations of these three parameters are expressed in Table 5. The W_LBP_ value in Bra A was smaller than that in Bra B, while the W_MBP_ value was greater. Compared with braless, the W_LBP_ value increased with the average difference of 1.87 cm in Bra A and 1.84 cm in Bra B due to the fabric thickness of bras. On the contrary, the W_MBP_ value decreased with an average difference of 0.79 cm and 1.13 cm in Bra A and Bra B, respectively. Moreover, the W_BB_ value in Bra B decreased with an average difference of 2.15 cm when compared with Bra A. The results show that Bra B had an even better effect on the gathering tendency of BP.

In addition, the improvement degree in breast gathering can also be evaluated by the R_BW_ value. Sun and Song [39] found that the R_BW_ value of a standard-breast was between 0.6 and 0.7. When the R_BW_ value is less than 0.6, the breast is considered to be gathering-breast; otherwise, the breast is classified as separating-breast. According to the results shown in Table 6, most subjects belonged to the gathering-breast in Bra A and Bra B, especially for Bra A. Majority of standard breasts were noted to be in braless condition.

#### 3.2.3. The Variation of the Breast Slice Area

The stereoscopic shape of the female breast influences the aesthetic evaluation as well. Females are expected to improve their stereoscopic shape after wearing a bra, such as making the breasts plumper and fuller. In this study, the breast-bra slice number and area (i.e., S and S_ij_) were used to represent the variations of breast-bra stereoscopic shape. The distribution of the S value is expressed in Table 7, which shows a range between three and eight for the three conditions.

Compared with braless, the S values in Bra A and Bra B were both concentrated in five and six, indicating that the breast-bra thickness increased and the breast shape showed a full and forward tendency after wearing a bra. Especially for Bra B, the S value of most subjects was six, showing a significantly increasing thickness of the breast-bra shape. However, for a few subjects with plump breasts who had S values of seven or eight when braless, the S value decreased in both Bra A and Bra B, indicating a slightly flattened trend of the breast-bra part after wearing the bras.

In Table 8, the S_ij_ values in slice one for all subjects were taken as an example to describe the variations of the breast-bra area. Compared with braless, the S_11_, S_12_, and S_14_ values all decreased in Bra A and Bra B, among which the S_11_ value decreased with the largest average difference of 7.30 cm^2^ in Bra A and 6.29 cm^2^ in Bra B. However, the S_13_ value increased with the average difference of 3.29 cm^2^ in Bra A and 2.27 cm^2^ in Bra B. Moreover, the total surface area of slice one showed decreasing trends with an average difference of 8.73 cm^2^ in Bra A, and 7.81 cm^2^ in Bra B.

### 3.3. Prediction of the Breast Morphology Parameters

After the analysis of morphology variations among the three conditions, L_BP_, W_LBP_, and G_BL_ can reflect the degree of breast ptosis, gathering, and the change of breast volume. To predict these three parameters after wearing Bra A and Bra B, 15 basic parameters (excluding the area parameters and the ratio parameters) were taken under nude condition to establish the prediction models. Since there were not simple linear relationships among these parameters, five methods (square, cube, and logarithms of three different bases) were tried to analyze the relationships. The plump oval type was taken as an example to describe the prediction process, and the methods were similar for the other types.

Stepwise regression analysis was used to establish a regression model for parameter prediction, and the regression equations for each of the morphological parameters are shown in Table 9. The error of each parameter was analyzed by comparing measured values with predicted values from 124 valid samples to verify the predictive accuracy of the regression models. The standard deviation, error range, and correlation coefficient between measurements and predictions for each parameter are shown in Table 10.

As for H_BP_, the adjusted R-square values were greater than 0.94, and the error ranges of 93.48% H_BP_A_ and 93.48% H_BP_B_ were all less than 2 cm. As for W_LBP_, although the adjusted R-square value was lower than other parameters, the error ranges of 95.65% W_LBP_A_ and 89.13% W_LBP_B_ were still less than 1 cm. However, the G_BL_ values of both bras had large ranges of error. The extreme G_BL_ value exceeded ±3 cm, with only 50.00% G_BL_A_ and 60.87% G_BL_B_ having an error range less than 1.5 cm. Therefore, the predicted models of G_BL_A_ and G_BL_B_ need to be improved. Since W and D were the main parameters to show the cross-section shape information, showing a high correlation with G_BL_, the G_BL_ could be predicted by the W__A_, W__B_, D__A_, and D__B_ values. The adjusted R-square values were greater than 0.95, and the error ranges of 91.30% G_BL_A_ and 91.30% G_BL_B_ were less than 1.5 cm. According to FZ/T 73012–2017, the permissible error of the under-band is ±2 cm, thus the error range of prediction models is acceptable.

The adjusted R-square values for all parameters are significant, indicating that the models are feasible. The correlation coefficients between measured value and predicted value are high, which shows a high agreement between predictions and measurements. The other two breast types can also be predicted by a regression model, but the parameters and coefficients used in the final prediction models are different.

## 4. Discussion

### 4.1. Comparison of Breast Ptosis under Three Conditions

The reduction in breast ptosis can be evaluated by the variations in the H_BP_ value. The most effective uplifting performance was denoted by Bra A, with an average value of 2.16 cm, while Bra B had an average value of 1.18 cm. Therefore, the degree of breast ptosis can be improved after wearing a bra with an upward trend in BP position. There may be two reasons for this situation. First, due to the different thicknesses of Bra A and Bra B, the direction of the force supporting the breasts may be different. With a thick cup pad (Bra B) at the lateral part of the bra, the breasts are subjected to componential forces both laterally and vertically, while there is almost only an upward force for Bra A. Second, the bra surface curve may influence the BP position, which is identified as the most convex point from the side view of the breast-bra part. However, the most convex point of Bra B is lower than that of Bra A, since the cup pad under the BP point is thicker, which causes Bra A to have a better performance in lifting.

### 4.2. Comparison of Breast Separating under Three Conditions

The improvement in breast separating can be evaluated by the variations of W_BB_ value. The performance in gathering effects of Bra B was superior to that of Bra A with average values of 2.15 and 1.72 cm, respectively. Therefore, the degree of the breast separating was improved after wearing a bra with an inward trend in BP position, especially for Bra B. Similarly, there may be two reasons for this situation. First, with a thicker cup at the lateral part of the breast wall for Bra B, there is an inward force to push the underneath soft tissues around the armpit into the bra cup. Second, the most convex points of Bra B are closer than those of Bra A, which caused Bra B to have a better performance in gathering.

### 4.3. Comparison of Breast-Bra Area under Three Conditions

The variation in breast-bra shape can be expressed via the surface area of the slices in four quadrants. Figure 5 shows the changes in breast-bra morphology after wearing a bra by taking one subject as an example. It can be seen that the breast-bra surface area of the second quadrant was the largest under the three conditions. The results also showed the highest position of the BP point, indicating that Bra A had a better performance in lifting the sagging breasts. While the surface areas of the first and second quadrants were both the largest, this indicates that Bra B was more effective in gathering the separating breasts to the center front. Under the braless condition, due to the gravity of the breast, the third and fourth quadrant areas of each slice were smaller than the first and second. The degree of breast ptosis and separation improved with the upward and inward moving trends of the BP when the force was supported by the bra. There were obvious effects on the degree of moving to the first quadrant after wearing both bras, which means that there was stronger support for gathering and upward lifting. As the under-breast was supported, the adipose and glandular tissue were redistributed.

However, there are several limitations to this study. The target subjects in this study are Chinese young females aging from 18 to 25 with cup sizes ranging from B to C. Although our sample was fairly diverse in breast morphology, it still could not reflect the whole Chinese young female population, which may be insufficient to achieve statistically accurate results. More research is needed to build a more comprehensive information base in order to recruit more females of all ages and cup sizes. It is recommended that more styles of bras be analyzed in the follow-up research to reveal the interaction between breasts and bras. Based on this research, the underlying relationship between breast morphology and specific bra features could help bra designers and manufacturers create a better fitting bra while meeting feminine aesthetic needs.

## 5. Conclusions

This paper proposed a “point-line-map-parameter” method based on 3D point cloud data of the human body to investigate and quantify the impact of bras with different thicknesses on the changes in breast shape. The reduction of breast ptosis, separation, and variation of the breast slice areas were used to assess the breast morphological changes before and after wearing a thin and thick cup bra. Combining parametric variation with breast aesthetic standards, the results showed that Bra A had a better performance in lifting the sagging breasts, reaching an average of 2.15 cm, and the improvement effect was mostly noted on the plump oval breast type (60.87%), while Bra B had a more significant impact on the appearance of the standard breast (47.83%). Furthermore, Bra B performed better in gathering the separating breasts to the center front by more than 2.15 cm laterally. The thick cup pad appears to be more effective at gathering the breasts together than lifting them up. It is probably the reason that the foam pad at the lateral part of the bra provides componential force both laterally and vertically, while there is almost only an upward force provided by the thin bra cup. To predict the variation of the breast shape with different bras, the calculation models of morphological parameters (H_BP_, W_LBP_, and G_BL_) were established for the plump oval type. Except for W_LBP_, the adjusted R-square of the regression equations were all greater than 0.7, and the prediction errors of all the parameters for about 90% of the subjects were within the range of 1–2 cm, showing that the regression model was feasible. This study can be applied to 3D modeling of the female breast, laying the groundwork for a visual presentation of variation with different types of bras for consumers.

## Figures and Tables

**Figure 1 ijerph-20-03856-f001:**
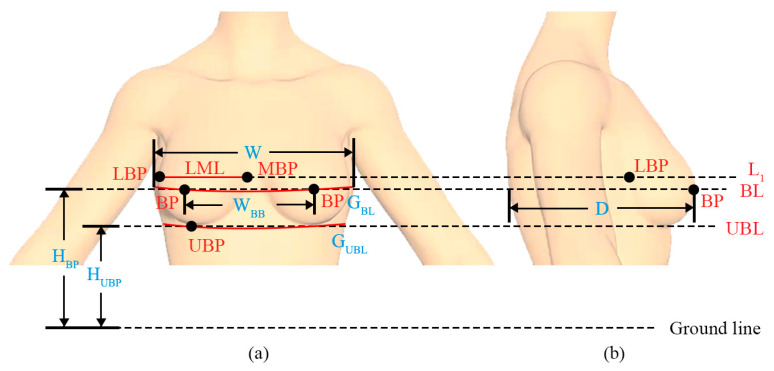
Legend of feature points and lines: (**a**) Body front view; (**b**) Body side view.

**Figure 2 ijerph-20-03856-f002:**
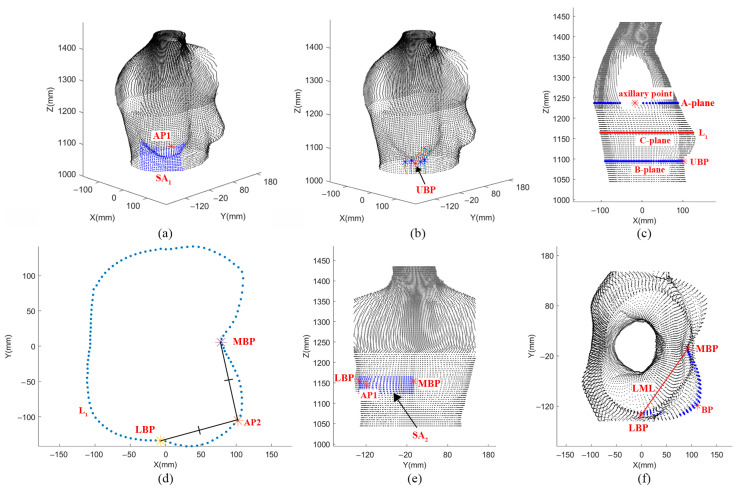
The implementing process of four feature points: (**a**) The search area of UBP; (**b**) The positioning of UBP; (**c**) The positioning of L_1_; (**d**) The positioning of MBP and LBP; (**e**) The search area of BP; (**f**) The positioning of BP.

**Figure 3 ijerph-20-03856-f003:**
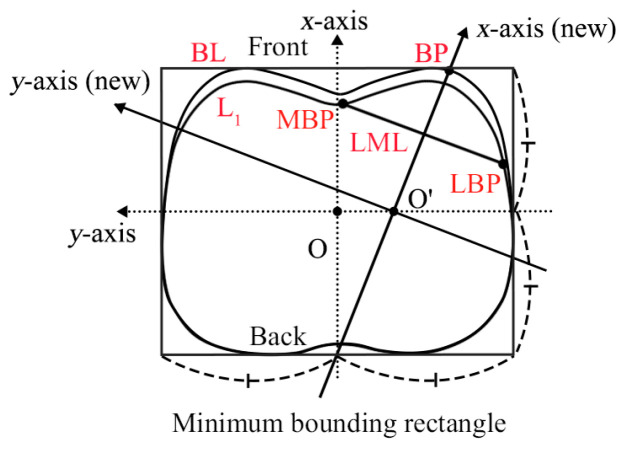
Schematic diagram of the local coordinate system.

**Figure 4 ijerph-20-03856-f004:**
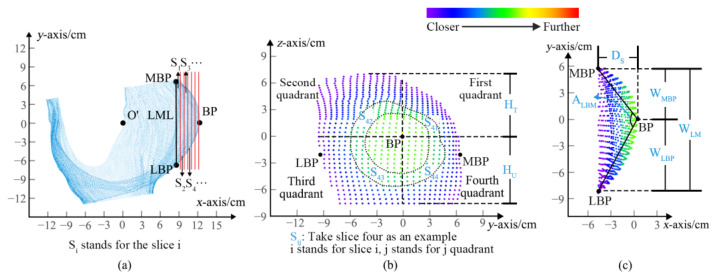
Schematic diagram of the breast slicing: (**a**) Top view of the right bust; (**b**) Front view of slices; (**c**) Top view of slices.

**Figure 5 ijerph-20-03856-f005:**
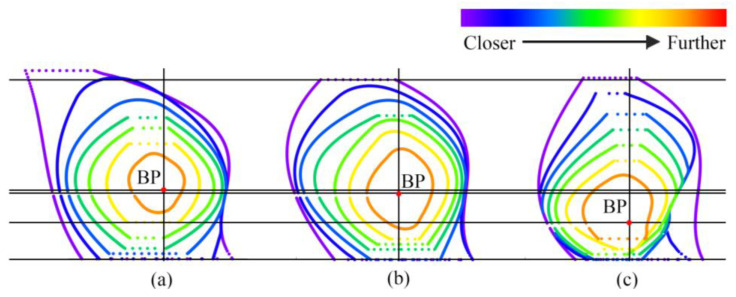
The change in quadrant area: (**a**) Bra A; (**b**) Bra B; (**c**) Braless.

**Table 1 ijerph-20-03856-t001:** Description of provided bras.

Types	Bra A (Thin Bra)	Bra B (Thick Bra)
Front view	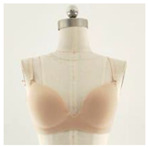	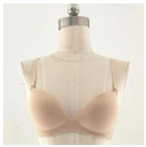
Top view	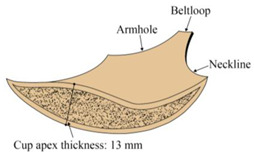	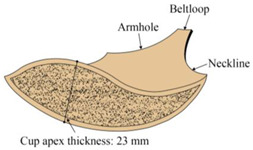
Cup apex thickness	13 mm	23 mm
Size	Uniform size	
Material	Mainly polyamide	
Description	With no underwire 3/4 cup Cups with pads and hooks Adjustable band at the under-bust	

**Table 2 ijerph-20-03856-t002:** Determination of feature points and lines.

Type	No.	Name	Definitions
Feature points	1	BP	Bust point, the most convex point of the breast
	2	MBP	Medial breast point, the medial-most endpoint of breast
	3	LBP	Lateral breast point, the lateral-most endpoint of breast
	4	UBP	Under-bust point, the lowest endpoint of breast
Feature lines	1	BL	Bust line, the cross-sectional curve passing through BP
	2	UBL	Under-bust line, the cross-sectional curve passing through LBP
	3	LML	The line passing through LBP and MBP
	4	L_1_	The cross-sectional curve passing through LBP
Feature parameters	1	A_LBM_	The angle that takes BP as the vertex, passing LBP and MBP
	2	D	Bust depth
	3	D_S_	The thickness of slices
	4	G_BL_	Bust girth
	5	G_UBL_	Under-bust girth
	6	H_BP_	The longitudinal distance between BP and the ground level
	7	H_UBP_	The longitudinal distance between UBP and ground level
	8	H	Body height
	9	H_T_	The longitudinal distance between O’ and the coordinate with the maximum z-value of slice five
	10	H_U_	The longitudinal distance between O’ and UBP
	11	W	Bust width
	12	W_BB_	The horizontal distance between the right BP and the left BP
	13	W_MBP_	The horizontal distance between MBP and O’
	14	W_LM_	The horizontal distance between LBP and MBP
	15	W_LBP_	The horizontal distance between LBP and O’
	16	R_BH_	The ratio of H_B_ to H
	17	R_BW_	The ratio of W_BB_ to W
	18	S	The number of slices
	19-n ^1^	S_ij_	The surface area of the slice i in the j quadrant

^1^ Note: n ranged between 31 and 51, varied with the number of slices.

**Table 3 ijerph-20-03856-t003:** The variations of H_T_, H_U_, and H_BP_ in different wearing conditions.

	Mean/cm			SD/cm			Range/cm		
	H_T_	H_U_	H_BP_	H_T_	H_U_	H_BP_	H_T_	H_U_	H_BP_
Bra A	7.24	7.42	116.07	1.28	1.20	5.16	4.25~11.50	5.00~11.00	100.01~130.84
Bra B	8.16	6.59	115.09	1.42	1.02	5.07	4.00~11.50	4.50~10.00	97.47~130.51
Braless	9.16	5.47	113.91	1.41	0.92	4.90	6.25~13.50	3.00~7.50	98.11~127.00

**Table 4 ijerph-20-03856-t004:** Distribution of the R_BH_ value for three types among three different conditions.

	Plump Oval Type			Uniform Oval Type			Flat Circle Type		
	Bra A	Bra B	Braless	Bra A	Bra B	Braless	Bra A	Bra B	Braless
Low- breast	5 (10.87%)	13 (28.26%)	28 (60.87%)	7 (14.00%)	11 (22.00%)	26 (52.00%)	0 (0.00%)	3 (10.71%)	9 (32.14%)
Standard- breast	13 (28.26%)	22 (47.83%)	12 (26.07%)	14 (28.00%)	20 (40.00%)	18 (36.00%)	4 (14.29%)	8 (28.57%)	9 (32.14%)
High- breast	28 (60.87%)	11 (23.91%)	6 (13.04%)	29 (58.00%)	19 (38.00%)	6 (12.00%)	24 (85.71%)	17 (60.71%)	10 (35.71%)
Total	46 (100.00%)			50 (100.00%)			28 (100.00%)		

**Table 5 ijerph-20-03856-t005:** The variations of W_LBP_, W_MBP_, and W_BB_ in different wearing conditions.

	Mean/cm			SD/cm			Range/cm		
	W_LBP_	W_MBP_	W_BB_	W_LBP_	W_MBP_	W_BB_	W_LBP_	W_MBP_	W_BB_
Bra A	8.25	6.39	16.69	0.72	0.74	1.44	6.58~10.43	4.14~7.98	11.91~20.23
Bra B	8.22	6.05	16.26	0.81	0.72	1.57	6.52~11.16	4.28~7.81	11.78~19.77
Braless	6.38	7.18	18.41	0.71	0.90	1.88	4.96~9.25	4.51~10.07	13.59~25.27

**Table 6 ijerph-20-03856-t006:** Distribution of the R_BW_ value for three types among three conditions.

	Plump Oval Type			Uniform Oval Type			Flat Circle Type		
	Bra A	Bra B	Braless	Bra A	Bra B	Braless	Bra A	Bra B	Braless
Gathering- breast	35 (76.09%)	30 (65.22%)	10 (21.74%)	41 (82.00%)	37 (74.00%)	6 (12.00%)	23 (82.14%)	20 (71.43%)	5 (17.86%)
Standard- breast	11 (23.91%)	16 (34.78%)	21 (45.65%)	9 (18.00%)	13 (26.00%)	33 (66.00%)	5 (17.86%)	8 (28.57%)	15 (53.57%)
Separating- breast	0 (0.00%)	0 (0.00%)	15 (32.61%)	0 (0.00%)	0 (0.00%)	11 (22.00%)	0 (0.00%)	0 (0.00%)	8 (28.57%)
Total	46 (100.00%)			50 (100.00%)			28 (100.00%)		

**Table 7 ijerph-20-03856-t007:** Distribution of the S value in different wearing conditions.

S	Bra A	Bra B	Braless
3	0	0	2
4	10	1	30
5	58	44	48
6	51	72	34
7	5	7	9
8	0	0	1
Total	124	124	124

**Table 8 ijerph-20-03856-t008:** The variations of S_11_, S_12_, S_13_, and S_14_ in different wearing conditions.

	Mean/cm^2^				SD/cm^2^				Range/cm^2^			
	S_11_	S_12_	S_13_	S_14_	S_11_	S_12_	S_13_	S_14_	S_11_	S_12_	S_13_	S_14_
Bra A	7.29	17.63	8.69	6.08	3.40	7.02	2.70	4.88	1.10~17.29	6.60~53.55	4.23~20.63	0.35~29.73
Bra B	8.30	19.60	7.67	5.31	2.71	7.69	2.68	3.54	1.84~16.68	4.56~49.71	3.08~17.97	0.07~18.12
Braless	14.59	21.32	5.40	7.11	4.01	7.95	2.27	3.59	5.95~28.81	9.12~49.43	1.58~13.54	0.11~16.93

**Table 9 ijerph-20-03856-t009:** Regression model of morphological parameters.

Morphological Parameters	Regression Equation	Adjusted R-Square	Sig.
H_BP_A_	$H_{BP\_A}=0.801+0.458\times \log_{2}\left(H_{\_C}\right)+1.242\times \mathrm{lg}\left(H_{UBP\_C}\right)$	0.949	0.000
H_BP_B_	$H_{BP\_B}=0.811+1.322\times \mathrm{lg}\left(H_{UBP\_C}\right)+0.608\times \ln\left(H_{\_C}\right)$	0.942	0.000
W_LBP_A_	$\begin{array}{ll} W_{L\mathrm{BP}\_A} & =-140.992+0.001\times W_{\_C}{}^{3}-1.140\times 10^{-5}\times A_{LBM\_C}{}^{3}\\ & +32.694\times \ln\left(A_{LBM\_C}\right)+2.771\times \ln\left(D_{\_C}\right) \end{array}$	0.681	0.000
W_LBP_B_	$W_{LBP\_B}=3.927+4.05\times 10^{-6}\times G_{BL\_C}{}^{3}+0.001\times W_{\mathrm{LM}\_C}{}^{3}$	0.511	0.000
G_BL_A_	$\begin{array}{ll} G_{BL\_A} & =-311.226+110.923\times \ln\left(G_{BL\_C}\right)-4.29\times 10^{-4}\times W_{BB\_C}{}^{3}\\ & +8.412\times 10^{-6}\times A_{\mathrm{LBM}\_C}{}^{3}+0.04\times H_{T\_C}{}^{3}-70.862\times \mathrm{lg}\left(W_{\_C}\right) \end{array}$	0.771	0.000
G_BL_B_	$\begin{array}{ll} G_{BL\_B} & =-155.316+0.002\times G_{BL\_C}{}^{2}+111.053\times \mathrm{lg}\left(A_{\mathrm{LBM}\_C}\right)\\ & -16.914\times \mathrm{lg}\left(H_{U\_C}\right)+0.001\times D_{\_C}{}^{3}+0.190\times D_{S\_C}{}^{2} \end{array}$	0.862	0.000
W__A_	$\begin{array}{ll} W_{\_A} & =-67.116+21.105\times \ln\left(G_{UBL\_C}\right)+0.026\times W_{\mathrm{LM}\_C}{}^{2}-0.007\times D_{S\_C}{}^{3}\\ & +21.125\times \mathrm{lg}\left(H_{BP\_C}\right) \end{array}$	0.762	0.000
W__B_	$\begin{array}{ll} W_{\_B} & =-27.705+2.41\times 10^{-4}\times W_{\_C}{}^{3}+3.34\times 10^{-4}\times D_{\_C}{}^{3}+4.393\\ & \times \log_{2}\left(W_{LM\_C}\right)-9.496\times \log_{2}(H_{U\_C})+21.408\times \mathrm{lg}(A_{\mathrm{LBM}\_C})\\ & +2.037\times H_{U\_C} \end{array}$	0.851	0.000
D__A_	$D_{\_A}=-30.943+10.215\times \log_{2}\left(D_{\_C}\right)+7.719\times \mathrm{lg}\left(W_{\_C}\right)$	0.823	0.000
D__B_	$\begin{array}{ll} D_{\_B} & =-41.767+12.631\times \log_{2}\left(D_{\_C}\right)+1.553\times 10^{-6}\times A_{\mathrm{LBM}\_C}{}^{3}\\ & +5.152\times \mathrm{lg}\left(W_{BB\_C}\right)+4.2\times 10^{-4}\times H_{T\_C}{}^{3} \end{array}$	0.898	0.000
G_BL_A_	$\begin{array}{ll} G_{BL\_A} & =11.832+1.775\times W_{\_A}+1.493\times D_{\_A}-2.316\times \log_{2}(H_{T\_C})\\ & -1.930\times H_{\mathrm{BP}\_C}{}^{3} \end{array}$	0.967	0.000
G_BL_B_	$G_{BL\_B}=8.780+1.502\times W_{\_B}+1.585\times D_{\_B}-0.003\times W_{LBP}{}^{3}$	0.959	0.000

Note: H_BP_A_ stands for the H_BP_ of Bra A; H_BP_B_ stands for the H_BP_ of Bra B; and H_BP_C_ stands for the H_BP_ of braless.

**Table 10 ijerph-20-03856-t010:** Error analysis of morphological parameters.

Morphological Parameters	Types	Mean/cm	Error Range/cm	R-Square
H_BP_A_	Measured value	115.33	2.10~2.67	0.975
	Predicted value	115.25		
H_BP_B_	Measured value	114.04	2.35~4.84	0.972
	Predicted value	114.02		
W_LBP_A_	Measured value	8.62	0.72~1.23	0.842
	Predicted value	8.62		
W_LBP_B_	Measured value	8.48	1.50~1.88	0.730
	Predicted value	8.48		
G_BL_A_	Measured value	91.13	1.94~2.37	0.985
	Predicted value	91.13		
G_BL_B_	Measured value	89.88	2.00~2.27	0.850
	Predicted value	89.88		

## Data Availability

Due to the sensitive nature of the questions asked in this study, survey respondents were assured that raw data would remain confidential and would not be shared.

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
