# Peer review of "Comparative Morphological Evaluation of Young Women’s Breast-Bra Reshaping by Different Bra Cups"

_ijerph, 2023, doi:10.3390/ijerph20053856_

Round 1
Reviewer 1 Report
See attached file

Reviewer 2 Report
The manuscript is interesting. The topic and findings enables bra designers and manufacturers to choose suitable bra cup features for different needs. However, some key issues must be properly addressed in details:
1. Grammatical and format mistakes: fullstop in L71, “study” instead of “studying” in L77
2. Please kindly elaborate FZ/T 73012-2008 in L87. It is different from the Standard FZ/T 73012-2017 (L53)?
3. Rephrase the sentences in Lines 91-93. “An” equation (in L93)
4. L94-95, please provide details of “previous studies”.
5. L99-100, please also give details of “our previous researches”.
6. BMI below 30 is a large range of body and breast shapes. As breasts are categorized by bra sizes, please also provide the distribution of bra sizes in this study. On the other hand, if human subject ethical approval has been granted for this study?
7. Table 2, it’s advised to provide the shape (cross-sectional views) of bra cups to better understand the influence of bra cups on the ultimate breast shape.
8. L342. What does it mean “Take Type 1 as an example”?
9. L350 R-square, instead of R2.
10. L355, L357, L361. Please rephrase. “There, which needs…”, “values were repredicted by them” “the feasible of the models”.
11. Please elaborate the limitation paragraph.
Round 2
Reviewer 1 Report
Please see attached file
